# Risk of second primary cancers after a diagnosis of first primary cancer: A pan-cancer analysis and Mendelian randomization study

Xiaohao Ruan[1†], Da Huang[1†], Yongle Zhan[2], Jingyi Huang[1], Jinlun Huang[1], Ada Tsui-Lin Ng[2], James Hok-Leung Tsu[2], Rong Na[2]*

[1]Department of Urology, Ruijin Hospital, Shanghai Jiao Tong University School of Medicine, Shanghai, China; [2]Division of Urology, Department of Surgery, Queen Mary Hospital, The University of Hong Kong, Hong Kong, China

**\*For correspondence:**
narong.hs@gmail.com

[†]These authors contributed equally to this work

**Competing interest:** The authors declare that no competing interests exist.

## Abstract

**Background:** The risk of second primary cancers (SPC) is increasing after the first primary cancers (FPC) are diagnosed and treated. The underlying causal relationship remains unclear.

**Methods:** We conducted a pan-cancer association (26 cancers) study in the Surveillance, Epidemiology, and End Results (SEER) database (non-Hispanic whites). The standardized incidence ratio (SIR) was estimated as the risk of SPCs in cancer survivors based on the incidence in the general population. Furthermore, the causal effect was evaluated by two-sample Mendelian Randomization (MR, 13 FPCs) in the UK Biobank (UKB, n=459,136,, European whites) and robust analysis (radial MR and Causal Analysis Using Summary Effect estimates, CAUSE).

**Results:** We found 11 significant cross-correlations among different cancers after harmonizing SIR and MR results. Whereas only 4 of them were confirmed by MR to have a robust causal relationship. In particular, patients initially diagnosed with oral pharyngeal cancer would have an increased risk of non-Hodgkin lymphoma (SIR$_{SEER}$ = 1.18, 95%Confidence Interval [CI]:1.05–1.31, OR$_{radial-MR}$=1.21, 95% CI:1.13–1.30, p=6.00 × 10$^{-3}$; OR$_{cause}$ = 1.17, 95% CI:1.05–1.31, p=8.90 × 10$^{-3}$). Meanwhile, ovary cancer was identified to be a risk factor for soft tissue cancer (SIR$_{SEER}$ = 1.72, 95%Confidence Interval [CI]:1.08–2.60, OR$_{radial-MR}$=1.39, 95% CI:1.22–1.58, p=1.07 × 10$^{-3}$; OR$_{cause}$ = 1.36, 95% CI:1.16–1.58, p=0.01). And kidney cancer was likely to cause the development of lung cancer (SIR$_{SEER}$ = 1.28, 95%Confidence Interval [CI]:1.22–1.35, OR$_{radial-MR}$=1.17, 95% CI:1.08–1.27, p=6.60 × 10$^{-3}$; OR$_{cause}$ = 1.16, 95% CI:1.02–1.31, p=0.05) and myeloma (SIR$_{SEER}$ = 1.54, 95%Confidence Interval [CI]:1.33–1.78, OR$_{radial-MR}$=1.72, 95% CI:1.21–2.45, p=0.02; OR$_{cause}$ = 1.49, 95% CI:1.04–2.34, p=0.02).

**Conclusions:** A certain type of primary cancer may cause another second primary cancer, and the profound mechanisms need to be studied in the future.

**Funding:** This work was in supported by grants from National Natural Science Foundation of China (Grant No. 81972645), Innovative research team of high-level local universities in Shanghai, Shanghai Youth Talent Support Program, intramural grant of The University of Hong Kong to Dr. Rong Na, and Shanghai Sailing Program (22YF1440500) to Dr. Da Huang.

## Editor's evaluation

This study presents a valuable finding on the causal relationship between second primary cancers and the initial diagnosis of a primary cancer via rigorous analysis of large-scale data. The strength of the study lies within the combination of pan-cancer analysis and the incorporation of Mendelian

randomization. The evidence supporting the claims of the authors is solid. The work will be of interest to clinicians and cancer biologists.

## Introduction

Cancer incidence is rapidly growing worldwide in the past decades. The reasons are complex including the aging of the population, the application of screening, environmental risk factors and genetic risk factors (*Sung et al., 2021*). In 2020, there were an estimated 19.3 million new cases and 10.0 million deaths of different types of cancers globally (*Sung et al., 2021*). In the US, there would be about 1.9 million new cases of cancers in 2023,and the most common type of cancers in the male and the female were prostate cancer (29%) and breast cancer (31%), respectively (*Siegel et al., 2022*; *Siegel et al., 2023*). Despite the rapidly increased incidence of cancers, the survival of (most types of) cancers in the US has vastly improved since the mid-1990s with medical advances and technical developments (*Siegel et al., 2022*; *Siegel et al., 2023*). For example, the 5-year relative survival rates of prostate cancer, melanoma, and female breast cancer, during 2011–2017 in the US were 98%, 93%, and 90%, respectively (*Siegel et al., 2022*). Such disparity may lead to an increase in the prevalence and the tumor burden in US society. More importantly, prolonged survival makes it possible for individuals to be diagnosed with a second primary cancer (SPC) after the first primary cancer (FPC) during the follow-up.

According to the Italian (1976–2010; *AIRTUM Working Group, 2013*), the Swiss (1981–2009; *Feller et al., 2020*), and the Swedish (1990–2015; *Zheng et al., 2020*) cancer registration data, increased risks of SPCs were observed in many types of cancer as the FPCs. Patients with oral cavity & pharynx, larynx, and esophagus as FPC were found to have a significantly elevated risk of any SPCs in both Italy and Switzerland (*Feller et al., 2020*; *AIRTUM Working Group, 2013*). In Sweden, liver cancers, as well as nasal and oral cancers, were found to be associated with SPCs (*Zheng et al., 2020*). In addition, several studies focused on certain cancer also suggested a potential relationship between FPC and SPC (*Chattopadhyay et al., 2018*). For example, increased FPC risk of colorectal cancer, kidney cancer, and melanoma were observed following the diagnosis of non-Hodgkin lymphoma. Despite the strong association observed in these studies, whether there is any underlying causal relationship is unknown, or the association observed is due to the potential confounders such as aging.

In the present study, our objectives are to perform a pan-cancer association study in the Surveillance, Epidemiology, and End Results (SEER) and to interpret the underlying causal relationship via Mendelian Randomization approaches using genetic variants in a large population cohort (UK Biobank, UKB). The study may help us understand critical questions in clinical practice about who should be more careful of the second primaries. In addition, precision screenings against certain cancers should also be considered among those patients with increased risk of second primaries in addition to the regular follow-up evaluations. In addition, many cancers are known to have a multifactorial etiology and some cancer treatments are known to be carcinogenic. It is essential to illustrate the potential relationship between FPCs and SPCs which mary optimize the treatment for patients' prognosis and survivorship.

## Methods
### Study populations

SEER Program 18 Registry database was obtained which covered 27.8% of the total population in the United States ("Number of Persons by Race and Hispanic Ethnicity for SEER Participants - SEER Registries.") (*National Cancer Institute, 2020*). The SEER program is the largest cancer incidence dataset in the United States based on population cancer registration. Based on the ICD-10 code, we identified adult patients (age ≥20 years) diagnosed with an FPC between 2000 and 2016, including 22 types of solid-tumor sites (oral cavity and pharynx, esophagus, stomach, small intestine, colon and rectum, liver, gallbladder, pancreas, larynx, lung and bronchus, bones and joints, soft tissue including heart, melanoma of the skin, female breast, male breast, cervix uteri, ovary, prostate, bladder, kidney, renal pelvis and ureter, brain, thyroid) and 4 types of hematological malignancies (Hodgkin lymphoma, non-Hodgkin lymphoma, myeloma, leukemia). Patients with diagnosis by autopsy or mentioned in the death certificate only were excluded.

**eLife digest** Better cancer treatment and early detection have increased survival rates among patients with cancer. But some cancer survivors can develop a second cancer called a second primary cancer. Second primary cancers may occur months or years after successful treatment of the primary cancer. They are not caused by the spread of the original tumor like a cancer metastasis. Instead, they appear to occur independently in another location or tissue.

Scientists are trying to understand what causes second primary cancers. Genetics, lifestyle, the environment, treatments used for the initial tumor, or other factors may all contribute to individuals developing a second cancer. Learning more about who is at risk of developing a second cancer and why, may lead to new prevention, treatment or screening strategies.

Ruan, Huang et al. found that people with some primary cancers have an increased risk of secondary primary cancers in specific tissues. The researchers first looked at the Surveillance, Epidemiology, and End Results (SEER) database that tracks US cancer patients to see if different types of cancers were more likely to lead to a second primary cancer. Then, the team conducted a comprehensive analysis for a causal relationship in a second extensive health database, the UK Biobank, to determine if the primary cancers may have caused the second primary cancer. The study showed that patients diagnosed with mouth or throat cancers were at increased risk of later developing a lymph node cancer called non-Hodgkin lymphoma. Patients diagnosed with ovarian cancer were at increased risk of later developing cancer in one of the body's soft tissues. Kidney cancer is likely the cause of later lung cancers and a type of blood cancer called myeloma.

Understanding the relationships between an initial and later cancer diagnosis is essential to improve cancer survivors' care. It is especially important for patients diagnosed early in life. More studies are needed to confirm the links Ruan, Huang et al. identified and to understand the mechanism. If more studies confirm the associations, physicians may want to screen survivors for specific cancers. Scientists may also be able to use the information to develop new strategies to help prevent or treat secondary primary cancers.

---

The UKB project is a prospective cohort study collecting phenotypic and genotypic data from ~500,000 individuals from across the United Kingdom (median follow-up time was ~14 years). The participants aged between 40 and 69 at recruitment (*Bycroft et al., 2018*). In the present study, a total of 459,156 participants with European Ancestry from UKB (release V3) with GWAS genotyping array data and imputation information were obtained and included in the MR analysis. Disease phenotypes in UKB were also defined using the ICD-10 code. Non-Caucasian patients were not included in the present study in SEER or UKB due to the small number of subjects in UKB, which made it hard to make the causal inference.

Written informed consent was obtained from all the participants from SEER or UKB according to the established standard of the studies. The study was approved by the Northwest Multi-centre Research Ethics in Manchester, UK (IRAS project ID: 299116; Application No. 66813).

## Genotyping and quality control

GWAS genotyping array data with imputation and QC from UK Biobank release V3 was obtained (*Bycroft et al., 2018*). Briefly, a total of 488,377 participants (after quality control, QC) were genotyped in UKB using two similar genotyping arrays, the UK Biobank Lung Exome Variant Evaluation (UK BiLEVE with 807,411 markers, n=49,950) and the Applied Biosystems UK BiLEVE Axiom Array by Affymetrix (825,927 markers, n=438,427). These two arrays share 95% of the markers. Individuals were excluded if: (a) ancestry testing using principal component analysis (PCA) to evaluate the potential conflicts between self-reported ethnicity/race and the genetic ethnicity/race; (b) Poor call rate at the genotyping stage (n=968, 0.2%); (c) Mismatched results between self-reported gender and genetic gender (n=652, 0.13%). The genotype concordance rate was reported as >99.0% (*Bycroft et al., 2018*). A total of 93,095,623 autosomal SNPs were identified in 487,442 individuals (*Bycroft et al., 2018*).

## Mendelian randomization

The conceptual framework was illustrated in *Figure 1* and the MR study was reported in accordance with the STROBE-MR guideline (*Skrivankova et al., 2021*). Two-sample MR analyses were performed

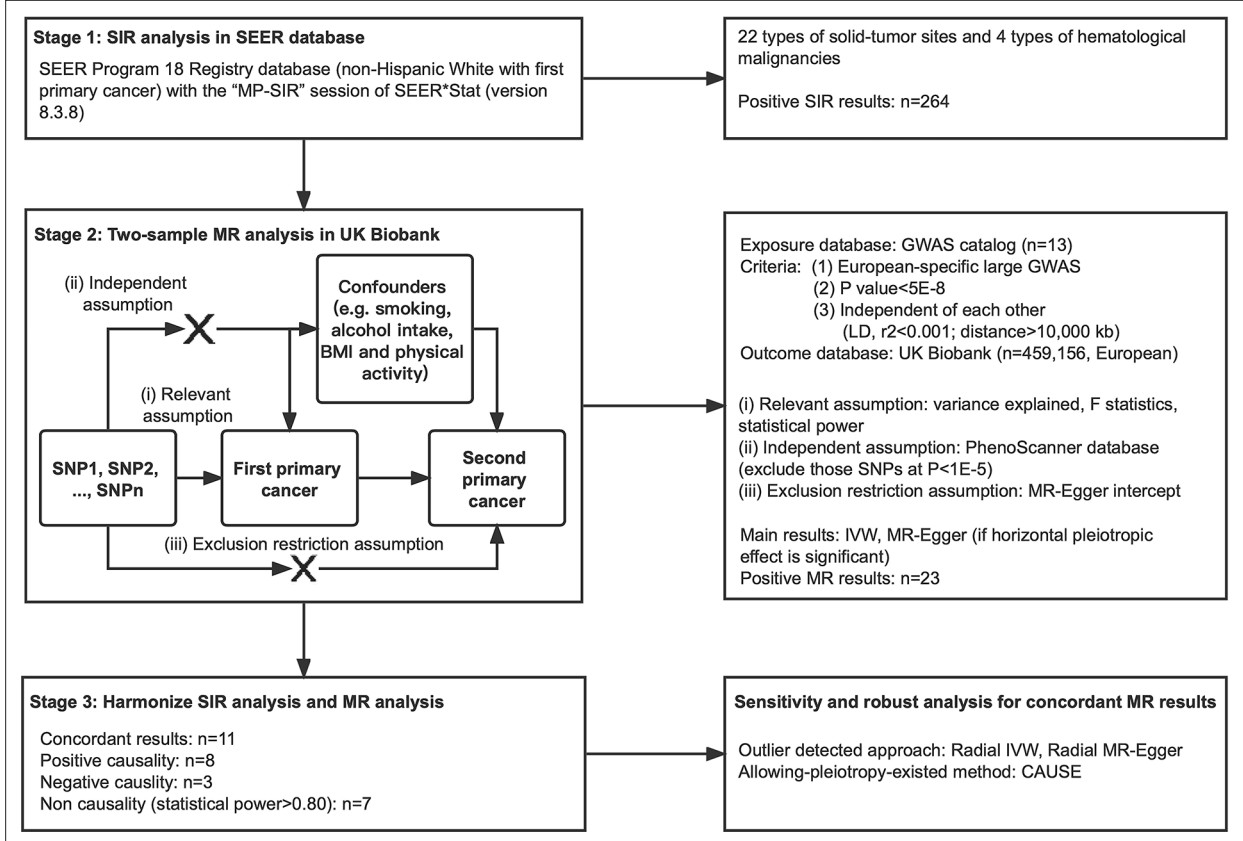

**Figure 1.** Study design and work flowchart.

The online version of this article includes the following figure supplement(s) for figure 1:

**Figure supplement 1.** Scatter plot.

**Figure supplement 2.** Funnel plot.

**Figure supplement 3.** Forrest plot.

**Figure supplement 4.** Leave-one-out plot.

to evaluate the causality between exposures (a certain primary cancer, FPC) and outcomes (another primary cancer, SPC). We used previously identified disease risk-associated SNPs from the GWAS Catalog database(https://www.ebi.ac.uk/gwas/) (**Buniello et al., 2019**). SNPs selection was based on the following criteria: (1) SNPs were from a single GWAS that identified the largest number of risk-associated SNPs and were conducted in European ancestry; (2) SNPs reached genome-wide significant level ($P<5 \times 10^{-8}$); (3) SNPs were independent of each other in terms of linkage disequilibrium (LD, $r^2 <0.001$) and distance (>10,000 kb).

MR analyses derive valid estimates where the following assumptions are met: (i) the SNPs are correlated with FPC, (ii) the SNPs affect SPC risk only through their effects on FPC and (iii) the SNPs are independent of any confounding factors for the association between FPC and SPC. For assumption (i), the strength of each instrument was measured using the F statistic and the proportion of the explained variance ($R^2$), which was considered to be sufficient if the corresponding F-statistic is >10. For assumption (ii) and (iii), we searched the PhenoScanner database (available at http://www.phenoscanner.medschl.cam.ac.uk/phenoscanner) (**Kamat et al., 2019**; **Staley et al., 2016**) to examine whether SNPs were significantly associated with established risk factors for certain cancers, including BMI, smoking, alcohol intake and physical inactivity, and excluded those at $p<1.0 \times 10^{-5}$. Statistical power calculations were performed using an online tool available at https://shiny.cnsgenomics.com/mRnd/(**Brion et al., 2013**). The statistical power was to capture an OR of 1.20 or 0.80 per a standard deviation (SD) change in the cancer risk.

Inverse-variance weighted MR (IVW-MR) and MR-Egger were used in the MR analyses (*Burgess et al., 2019*; *Burgess and Thompson, 2017*; *Davies et al., 2018*; *Hemani et al., 2018*). MR would be performed based on at least 4 SNPs. Briefly, these two methods are the most used MR methods to infer a causal relationship. IVW-MR is based on a random effect model and is the most efficient with the greatest statistical power (*Hemani et al., 2018*). Potential bias as horizontal pleiotropy was evaluated and adjusted via MR-Egger (*Burgess et al., 2019*; *Burgess and Thompson, 2017*; *Davies et al., 2018*; *Hemani et al., 2018*). The causal inference was interpreted via IVW-MR results if the horizontal pleiotropic effect was not significant; otherwise, based on MR-Egger. A causal relationship will only be interpreted when a significant MR result was observed from MR analyses together with a significant association based on the SEER database. A series of sensitivity and robust analyses including leave-one-out, Radial MR (*Bowden et al., 2019*; *Bowden et al., 2018*) and CAUSE (*Morrison et al., 2020*) methods would be performed in case of concordant significant results. MR analyses were performed using the R package 'TwoSampleMR', 'MendelianRandomization', 'RadialMR' and 'cause'.

## Statistical analysis

The multiple primary standardized incidence ratios (MP-SIR) were defined as the observed incidence of a second malignancy among cases previously diagnosed with a certain type of cancer divided by the expected incidence based on the SEER referent population (the SEER18 2000–2016 referent rate file). All the standardized incidence observed/expected (O/E) ratios (SIR) and the corresponding 95% confidence intervals (95% CI) were derived using the 'MP-SIR' session of SEER*Stat (version

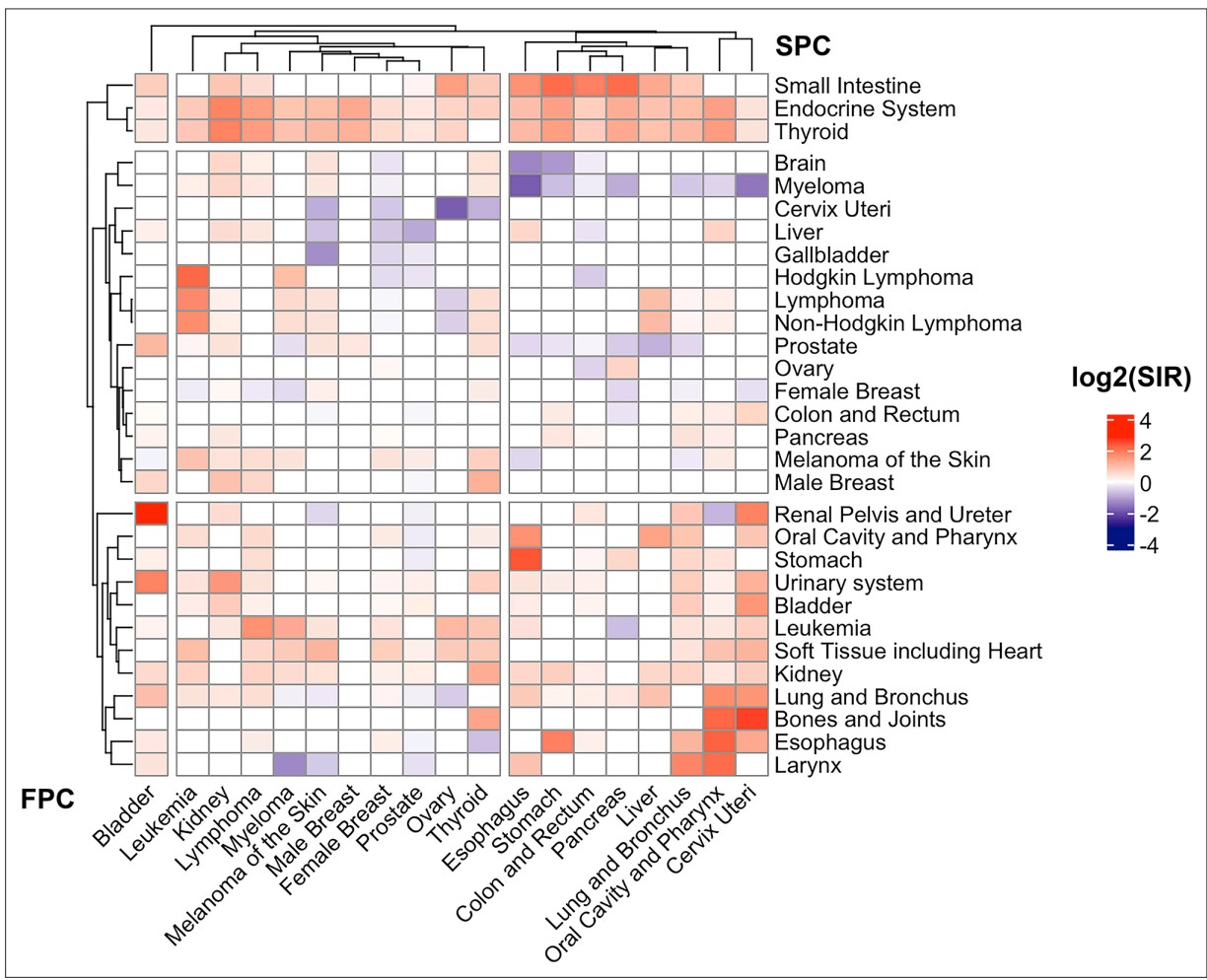

**Figure 2.** Heat-map of standard incidence ratio for First primary cancers (FPC, horizontal, cancer listed in the bottom) and second primary cancers (SPC, vertical, cancer listed on the right) in the SEER 18 registry (2000–2016). The standardized incidence observed/expected (O/E) ratios and the corresponding 95% confidence intervals were derived using the "MP-SIR" session of SEER*stat (version 8.3.8).

8.3.8)("Multiple Primary - Standardized Incidence Ratios - SEER*Stat.") (**National Cancer Institute, 2023**). Only the non-Hispanic white population was included in the present study. We restricted a minimum 2-month latency period between the first and second primary diagnosis (**Davis et al., 2014**). Subgroup analyses were performed after stratifying by radiation therapy (received or not). All statistical analyses were performed using SEER*STAT and R software (4.1.2) (**R Development Core Team, 2021**). A type I error of 0.05 (two-sided) was used to define statistical significance. Multiplicity effects were only considered during the selection of SNPs.

## Results

We set out to identify the observational association between FPCs and SPCs among 22 solid tumors and 4 hematological malignancies based on the SEER database (**Figure 1**). A total of 264 significant standardized incidence ratio (SIR) existed among them when compared to a standard population (**Supplementary file 1a-1q**). Hierarchical clustering analysis (heatmap) is shown in **Figure 2** which illustrates the comparison between the cancer incidence in patients with a certain type of FPC and the incidence in the population average level in the SEER dataset. Second primary thyroid cancer, small bowel cancer, or cancers of endocrine system were strongly and significantly associated with most of the FPCs. Cancers of the digestive system, cervix uteri, as well as lung cancer were clustered together. They are closely correlated with an increased risk of several types of cancers such as bladder cancer, kidney cancer, etc. Our subgroup analysis suggested that patients with prostate cancer who received radiation therapy were at an increased risk of being diagnosed with another type of primary cancer during the follow-up including small intestine, soft tissue, and leukemia, compared with those

**Table 1.** Summary of the cancer-specific instrument variables used in this study (European ancestry).

| Cancer type | GWAS Source | PMID | Number of SNPs* | Cases | Total population | Variance explained, $R^2$, % | F-statistics |
|---|---|---|---|---|---|---|---|
| Oral and pharynx | GCST003857 | 27749845 | 7 | 6034 | 12,619 | 2.22 | 283.68 |
| Larynx | GCST010285 | 32276964 | 1[†] | 394 | 4887 | 0.28 | 13.72 |
| Esophagus | GCST003740 | 27527254 | 5 | 10,279 | 27,438 | 0.72 | 198.94 |
| Stomach | GCST002990 | 26098866 | 1[†] | 2043 | 202,533 | 0.02 | 34.44 |
| Colon and rectum | GCST003017 | 26151821 | 8 | 18,299 | 37,955 | 0.89 | 340.75 |
| Pancreas | GCST005434 | 29422604 | 14 | 9040 | 21,536 | 4.28 | 962.28 |
| Melanoma | GCST004142 | 28212542 | 17 | 6628 | 293,193 | 0.29 | 852.68 |
| Lung | GCST004748 | 28604730 | 7 | 29,266 | 85,716 | 0.55 | 474.00 |
| Male Breast | GCST011526 | 32785646 | 2[†] | 2190 | 6836 | 1.27 | 87.91 |
| Female Breast | GCST004988 | 29059683 | 104 | 122,977 | 228,951 | 4.95 | 11917.81 |
| Cervix uteri | GCST004833 | 28806749 | 1[†] | 2866 | 9347 | 0.45 | 42.24 |
| Ovary | GCST002748 | 25581431 | 9 | 18,530 | 69,745 | 1.16 | 818.42 |
| Prostate | GCST006085 | 29892016 | 71 | 79,148 | 140,254 | 5.47 | 8111.66 |
| Bladder | GCST002240 | 24163127 | 7 | 2305 | 6206 | 8.46 | 572.81 |
| Kidney | GCST004710 | 28598434 | 8 | 10,784 | 31,190 | 1.11 | 349.99 |
| Thyroid | GCST004144 | 28195142 | 6 | 3001 | 290,551 | 0.10 | 290.83 |
| Myeloma | GCST004483 | 27363682 | 13 | 9866 | 249,054 | 0.31 | 774.24 |

Note: See **Supplementary file 2a**: the list of SNPs included in the final calculation for each phenotype.

No study found in European: bone and joint, brain, liver, small intestine, gallbladder, renal pelvis and ureter.

Heterogeneity: lymphoma, leukemia.

Too few SNPs: larynx, stomach, male breast cancer.

*Number of SNPs included in the final calculation of PRS in our study; not necessarily the total number of SNPs from the source due to the filtering steps discussed in the main text and germline data availability.

[†]MR would be performed based on at least 4 SNPs.

**Table 2.** Concordant causality between Mendelian randomization results and SEER analysis.

| Cancer type (first primary cancer) | Cancer type (second primary cancer) | | |
| --- | --- | --- | --- |
| | Positive Causality | Negative Causality | No Causality (statistical power ≥0.80) |
| Oral and pharynx | Non-hodgkin lymphoma | - | Female breast, Prostate |
| Esophagus | - | - | - |
| Colon and rectum | - | - | - |
| Pancreas | Small intestin | - | Melanoma |
| Melanoma | Female breast | - | Colon and Rectum |
| Lung | - | Female Breast | - |
| Female Breast | - | - | - |
| Ovary | Soft tissue | - | - |
| Prostate | - | Colon and Rectum | Non-Hodgkin lymphoma |
| Bladder | - | - | Female breast, Non-hodgkin lymphoma |
| Kidney | Lung and Bronchus, Melanoma, Non-hodgkin lymphoma, Myeloma | - | - |
| Thyroid | - | - | - |
| Myeloma | - | Lung and Bronchus | - |

See *Supplementary file 2c* for unconcordant causality result; *Supplementary file 2d-p* for the details of Mendelian randomization results (IVW and MR-Egger).

without such therapy (*Supplementary file 1m*). And SPC risk after breast cancer was inconsistent among males and females. Men with breast cancer had a higher risk for thyroid cancer and prostate cancer ($SIR_{SEER}$ = 1.30; 95% CI, 1.13–1.49; $SIR_{SEER}$ = 2.33, 1.12–4.29; *Supplementary file 1j*), but with no significant risk change for other cancers.

Details of the included exposure-associated SNPs in European ancestry were shown in *Table 1* and *Supplementary file 2a*. The number of SNPs ranged from 5 (esophagus) to 104 (female breast), and the proportion of variance explained by SNPs ($R^2$) ranged from 0.10% (thyroid) to 8.46% (bladder). F-statistics for all 13 cancers exceeded 10, suggesting no weak instrument bias here. However, some problems, including too few SNPs, no related GWAS and disease heterogeneity, affected the comprehensive MR analysis (*Table 1*). Due to the low incidence of cancers in UKB, the power to detect a significant effective size (0.8/1.2) was relatively low, except for colorectal cancer, lung cancer, female breast cancer and prostate cancer (*Supplementary file 2b*).

Results from MR analyses are presented in *Supplementary file 2c-2p*. A total of 23 significant association was detected (16 positive causality and 7 negative causality). Concordant significant results and unconcordant results between MR and the SEER SIR analyses are shown in *Table 2* and *Supplementary file 2c*, respectively. The concordant results suggested that patients diagnosed with primary oral and pharynx cancer would cause a significantly increased risk of second primary non-Hodgkin lymphoma ($SIR_{SEER}$ = 1.18, 95% CI: 1.05–1.31; IVW-MR $P$=8.96 × 10$^{-4}$). After a primary diagnosis of pancreatic cancer, SPC risks were increased for small intestine ($SIR_{SEER}$ = 4.37, 95% CI: 2.85–6.40; MR-Egger p=0.04). It also indicated that female patients initially diagnosed with melanoma would cause a mild but significantly increased risk of cancers in the breast ($SIR_{SEER}$ = 1.17, 95% CI: 1.12–1.23; IVW-MR p=0.04), as well as ovary cancer on soft tissue related cancer ($SIR_{SEER}$ = 1.72, 95% CI: 1.08–2.60; IVW-MR p=8.39 × 10$^{-5}$). The greatest number of casual relationships were observed in kidney cancer as FPC. A primary kidney cancer might cause an elevated risk of cancers of lung and bronchus ($SIR_{SEER}$ = 1.28, 95% CI: 1.22–1.35; IVW-MR p=0.01), non-Hodgkin lymphoma ($SIR_{SEER}$ = 1.19, 95% CI: 1.08–1.31; IVW-MR p=3.64 × 10$^{-3}$), myeloma ($SIR_{SEER}$ = 1.54, 95% CI: 1.33–1.78; IVW-MR p=3.94 × 10$^{-3}$). Meanwhile, some primary cancer site might give protective effect against another cancer (for

**Table 3.** Sensitivity and robust analysis of the concordant causality with outlier-filtering approaches.

| | Radial IVW | | Radial MR-Egger | | CAUSE | |
|---|---|---|---|---|---|---|
| Exp-out | OR (95% CI) | p | OR (95% CI) | p | OR (95% CI) | p |
| Oropharynx-NHL | 1.21 (1.13–1.30) | $6.00\times10^{-3}$ | 1.21 (1.13–1.30) | 0.52 | 1.17 (1.05–1.31) | $8.90\times10^{-3}$ |
| Pancreas-Intestin | 1.05 (0.83–1.32) | 0.69 | 2.39 (0.91–6.31) | 0.10 | 1.03 (0.79–1.35) | 1.00 |
| Melanoma-Breast | 1.08 (1.00–1.15) | 0.06 | 1.05 (0.84–1.32) | 0.69 | 1.04 (0.95–1.16) | 0.75 |
| Lung-Breast | 0.86 (0.79–0.93) | $9.76\times10^{-3}$ | 0.77 (0.54–1.08) | 0.19 | 0.89 (0.76–1.04) | 0.23 |
| Ovary-Soft | 1.39 (1.22–1.58) | $1.07\times10^{-3}$ | 1.3 (0.74–2.29) | 0.39 | 1.36 (1.16–1.58) | 0.01 |
| Prostate-CRC | 1.00 (0.95–1.04) | 0.94 | 0.88 (0.80–0.98) | 0.02 | 0.99 (0.93–1.04) | 0.99 |
| Kidney-Lung | 1.17 (1.08–1.27) | $6.60\times10^{-3}$ | 0.94 (0.62–1.43) | 0.78 | 1.16 (1.02–1.31) | 0.05 |
| Kidney-Melanoma | 1.33 (1.02–1.73) | 0.04 | 0.56 (0.18–1.80) | 0.37 | 1.25 (0.96–1.73) | 0.51 |
| Kidney-NHL | 1.25 (1.11–1.40) | $7.87\times10^{-3}$ | 1.33 (0.72–2.46) | 0.39 | 1.20 (0.99–1.43) | 0.09 |
| Kidney-Myeloma | 1.72 (1.21–2.45) | 0.02 | 0.43 (0.11–1.77) | 0.29 | 1.49 (1.04–2.34) | 0.02 |
| Myeloma-Lung | 0.92 (0.86–0.98) | 0.02 | 1.09 (0.79–1.51) | 0.61 | 0.93 (0.86–1.00) | 0.21 |

IVW = Inverse variance weighted. CAUSE = Causal Analysis Using Summary Effect estimates. OR = odds ratio. CI = confidence interval. exp = exposure. out = outcome. NHL = non-Hodgkin lymphoma. CRC = colorectal cancer.

instance, lung cancer vs. female breast cancer, *Table 2*). Scatter plot, Funnel plot, forest plot and leave-one-out analysis showed single SNP effective size in *Figure 1—figure supplements 1–4*, respectively.

More conservative analyses were performed to further confirm these causal relationships. We applied 2 outlier-detected methods with modified second order weights (radial IVW and radial MR-Egger) and CAUSE (Causal Analysis Using Summary Effect estimates) to each pair of phenotypes, with the rationale that robust relationships would exhibit consistent and statistically significant results across different methods. Additionally, CAUSE is the only method capable of distinguishing causality from both correlated and uncorrelated pleiotropy. The relationship with at least two significant results was treated as a robust causality (*Table 3*). We found consistent evidence for a causal effect of oral and pharynx cancer on non-Hodgkin lymphoma ($P_{radial-IVW}$=$6.00 \times 10^{-3}$, $P_{cause}$ = $8.90 \times 10^{-3}$), ovary cancer on soft tissue cancer ($P_{radial-IVW}$=$1.07 \times 10^{-3}$, $P_{cause}$ = 0.01), kidney cancer on lung and myeloma ($P_{radial-IVW}$=$6.60 \times 10^{-3}$, $P_{cause}$ = 0.05; $P_{radial-IVW}$=0.02, $P_{cause}$ = 0.02).

## Discussion

With the expanded life expectancy and the prolonged survival of cancers, the incidence of SPCs has been rapidly growing in the past decades (*Copur et al., 2019*). Genetic factors or shared environmental factors are probably the major causes. In the previous association studies, individuals with a certain type of primary malignancy would have an increased risk of another malignancy (*Feller et al., 2020*; *AIRTUM Working Group, 2013*; *Zheng et al., 2020*). However, whether there are any causal effects within the associations is unclear. In the present study, via the association study based on the SEER database and the MR approach using the UKB genetic dataset, we were able to perform this comprehensive investigation across 26 different types of cancers. 13 out of 26 types of cancers with adequate GWAS data were able to be further investigated using MR analysis. We found that numbers of primary malignancies were associated with an increased risk of a second primary malignancy, however, only a small part of the associations would have a causal relationship (*Table 2*).

Many significant findings were observed in the SEER SIR analysis. SEER is one of the largest cancer registration-based datasets making itself the most proper data source to answer the study objectives; however, several advantages of this database and limitations of the results should be noted. First, the results from the SEER SIR analysis were associations rather than causal inferences. Many factors may influence the results of the associations. For example, the confounder of screening effects may exist. Patients diagnosed with primary cancer might have more frequent healthcare visits compared to those who did not have any cancers. Therefore, some indolent cancers such as thyroid cancer and low-risk prostate cancer could have been over-diagnosed due to the screening effects. Second, it is a

cancer registration-based cohort rather than a population cohort, the standardized incidences calculated from the SEER database may not represent the situation in the general population. Third, treatment preferences and follow-up strategies may vary in different locations or institutions, which would also affect the occurrence/detection of the second primary cancer. For example, radiation therapy may increase the risk of cancer in nearby organs[6]. However, treatment and follow-up information are not completed in the SEER database due to the natural design of the cohort[31]. More importantly, lifestyles, comorbidities, and environmental factors were not included in the SEER database. These are important confounders of the associations between the first primary cancers and the second primary cancers.

The MR approaches in the current study revealed some interesting findings, but several non-concordant results between MR analyses and SEER SIR were also observed (Table S20). It does not indicate that the causal relationship does not exist. Some factors, such as the period of follow-up, may conceal and cause false negatives in the association study (SEER SIR). For example, pancreatic cancer might cause an increased risk of cancers in esophagus, colon and rectum, etc. based on the MR analysis in our study; however, no association was observed in SEER SIR analysis. The short and poor survival of pancreatic cancer could be the most critical reason for the failure of finding a positive association in the population data --- simply did not have enough time of follow-up to observe the outcomes. Therefore, the interpretation of these results should be more careful at this stage.

The lack of GWAS findings would be a major limitation of the MR approach for some diseases as in the present study. The MR approach may only represent part of the biological effects in the causal pathway between the exposure and the outcome. A final causal inference should always be established based on biological mechanisms. From the angle of organ location, some cancers (ovary cancer and soft tissue sarcoma) might share the same tumor-related or tumor-developing environment. Besides the outside therapeutic settings (radio- and chemotherapy, immuno-suppressive agents) and individual factors (smoking, hormone level, certain occupational settings, HIV or HPV infections, and family histories), FPC might also influence the iatrogenic immune by suppressing antitumor defense mechanisms via inflammation or other meditating effects (*Shalapour and Karin, 2019*). For instance, the increased risk of renal cell cancer, and non-Hodgkin lymphoma were observed in immunosuppressed patients in Denmark and Sweden (*Hortlund et al., 2017*). The immune factor might be the inner relation between FPC and SPC. In our assumption, the question of mechanism should be answered via cross-trait GWAS meta-analyses, searching for shared genetic architecture under high heritability or potential meditation factors with comprehensive database and analyses, functional experiments (tissue- or cell-specific findings) and final validation in a cohort of comorbidity patients. In terms of clinical implications, lifestyle, environment, treatment, host factors and other influences may contribute to multiple primary cancer. It required clinicians and researchers to explore a specific link between FPCs and SPCs. It would be of clinical importance to make a personalized screening plan for certain primary cancer patients (eg. oral pharyngeal cancer, ovary cancer and kidney cancer), especially for those young onsets. For example, patients with kidney cancer would have an increased risk of lung cancer and myeloma. Thus, a screening for lung cancer and myeloma should be recommended among those patients.

Finally, besides the limitations mentioned, the relatively small number of cases of some diseases in UKB may lower our statistical power. As UKB is a population-based prospective cohort, the relatively short follow-up period may not allow us to observe enough events (multiple cancers) at this stage. And it is expected to be independently replicated in another dataset. Regardless, these findings progress our understanding of the relationship underlying both FPCs and SPCs, and potentially provide points of exploration and intervention that may reduce second primary cancers.

## Conclusion

Patients who were diagnosed with a certain type of primary cancer may cause another type of primary cancer, especially pharynx cancer on non-Hodgkin lymphoma, ovary cancer on soft tissue cancer, kidney cancer on lung and myeloma. The profound mechanisms need to be studied in the future.

## Acknowledgements

The current study contains a part of the capstone thesis by Rong Na for Master of Public Health at Johns Hopkins University School of Public Health (the results were updated based on the updated

datasets); otherwise, the study content has never been published elsewhere. We would like to thank for the supervision from Professor Jianfeng Xu from NorthShore University Health System, IL USA, Professor William B Issacs and Professor Bruce J Trock from Johns Hopkins University, during Rong Na's capstone projects. We thank SEER program to approve our protocol and provide the custom datasets. We also thank the UK Biobank for access to the data (Project Number: 66813). Funding information This work was in supported by grants from National Natural Science Foundation of China (Grant No. 81972645), Innovative research team of high-level local universities in Shanghai, Shanghai Youth Talent Support Program, intramural grant of The University of Hong Kong to Dr. Rong Na, and Shanghai Sailing Program (22YF1440500) to Dr. Da Huang. All the funders had no role in study design, data collection, data analysis, interpretation, and writing of the report.

## Additional information

### Funding

| Funder | Grant reference number | Author |
|---|---|---|
| National Natural Science Foundation of China | 81972645 | Rong Na |
| Innovative Research Team of High-level Local University in Shanghai | | Rong Na |
| Shanghai Youth Talent Support Program | | Rong Na |
| University of Hong Kong | Intramural grant | Rong Na |
| Shanghai Sailing Program | 22YF1440500 | Da Huang |

The funders had no role in study design, data collection and interpretation, or the decision to submit the work for publication.

### Author contributions

Xiaohao Ruan, Software, Formal analysis, Visualization, Methodology, Writing - original draft, Writing - review and editing; Da Huang, Resources, Software, Formal analysis, Funding acquisition, Visualization; Yongle Zhan, Software, Methodology; Jingyi Huang, Jinlun Huang, Data curation, Investigation; Ada Tsui-Lin Ng, James Hok-Leung Tsu, Project administration; Rong Na, Conceptualization, Resources, Data curation, Supervision

### Author ORCIDs

Xiaohao Ruan http://orcid.org/0000-0001-8373-5237
Da Huang http://orcid.org/0000-0002-6203-9459
Rong Na http://orcid.org/0000-0001-7470-5108

### Ethics

Written informed consent was obtained from all the participants from SEER or UKB according to the established standard of the studies. The study was approved by the Northwest Multi-centre Research Ethics in Manchester, UK (IRAS project ID: 299116; Application No. 66813).

### Decision letter and Author response

Decision letter https://doi.org/10.7554/eLife.86379.sa1
Author response https://doi.org/10.7554/eLife.86379.sa2

## Additional files

### Supplementary files

• Supplementary file 1. Standard incidence ratio analysis of different types of cancers in patients with malignancies in SEER database.

• Supplementary file 2. Mendelian randomization results of FPCs and SPCs in UK Biobank database.

• MDAR checklist

## Data availability

Data used in this research are publicly available to qualified researchers on application to GWAS catalog (https://www.ebi.ac.uk/gwas/), the SEER database (https://seer.cancer.gov/) and UK Biobank (https://www.ukbiobank.ac.uk/). Source code for all analyses were in related software or R packages.

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
