## [Editor Report]

This study presents a valuable finding on the causal relationship between second primary cancers and the initial diagnosis of a primary cancer via rigorous analysis of large-scale data. The strength of the study lies within the combination of pan-cancer analysis and the incorporation of Mendelian randomization. The evidence supporting the claims of the authors is solid. The work will be of interest to clinicians and cancer biologists.

---

## [Decision Letter]

**Decision letter after peer review:**

Thank you for submitting your article "Risk of Second Primary Cancers After a Diagnosis of First Primary Cancer: A Pan-Cancer Analysis and Mendelian Randomization Study" for consideration by *eLife*. Your article has been reviewed by 2 peer reviewers, and the evaluation has been overseen by a Reviewing Editor and Caigang Liu as the Senior Editor.

Essential revisions:

1) First primary ovarian cancers can be classified into different subtypes to further analyze the relationship between subtypes and causal secondary primary cancers, such that secondary primary cancer prevention would be more specific and targeted.

2) Please consider providing a better rationale for why studying second primary cancers is crucial.

3) Please provide a discussion on the potential clinical implications of the observed associations between first and second primary cancers.

*Reviewer #1 (Recommendations for the authors):*

1. Four prominent groups of causal relationships mentioned in the research could be further explored. For instance, first primary ovary cancers can be classified into different subtypes to further analyze the relationship between subtypes and causal secondary primary cancers, so that secondary primary cancer prevention would be more specific and targeted.

2. The limited SNPs included in some cancer types mentioned in the research might lead to missed opportunities to discover other significant associations. Enlargement of the data source could have benefits.

3. SEER database used in stage 1 and UK Biobank database used in stage 2 are based on completely different populations, and this may cause confounding factors due to genetic heterogeneity, which should be considered.

*Reviewer #2 (Recommendations for the authors):*

The paper investigates the association and causal relationship of second primary cancers following a diagnosis of first primary cancer. The combination of a pan-cancer analysis and the use of Mendelian randomization adds novelty and value to the study. However, as the author state, the objective of the study is to investigate the causal relationship between FPCs and SPCs. This can be done based on the UKB data. it is not clear why use SEER data which only provides the association.

---

## [Author Response]

Essential revisions:1) First primary ovarian cancers can be classified into different subtypes to further analyze the relationship between subtypes and causal secondary primary cancers, such that secondary primary cancer prevention would be more specific and targeted.

Thank you for your comments. We performed subgroup analyses in SEER Research Plus Data 18 Reg, Nov 2020 Sub (2000-2018), with additional 2-year follow-up data. We stratified the total patients with first primary ovarian cancers to four groups based on the pathological subtypes: serous (~69.5%), endometrioid, mucinous, and clear cell carcinomas. The high-grade serous carcinoma (HGSC), as the most common type of ovarian cancer, was also evaluated. The SIR analyses were repeated in the above subgroups. The additional result was listed in Supplementary file 1l. However, only some subtypes (e.g. serous, high-grade serous and endometrioid ovary cancer) conformed to the results in the relation between overall primary ovary cancer and several secondary primary cancers (lung and bronchus and non-Hodgkin lymphoma).

2) Please consider providing a better rationale for why studying second primary cancers is crucial.

We’ve added it on lines 90-97. The study may help us understand critical questions in clinical practice about who should be more carefully monitored for the second primaries. In addition, precision screenings against certain cancers should also be considered among those patients with increased risk of second primaries in addition to the regular follow-up evaluations. As known, many cancers are known to have a multifactorial etiology and some cancer treatments are known to be carcinogenic. It is essential to illustrate the potential relationship between FPCs and SPCs which may optimize the treatment for patients’ prognosis and survivorship.

3) Please provide a discussion on the potential clinical implications of the observed associations between first and second primary cancers.

In terms of clinical implications, lifestyle, environment, treatment, host factors and other influences may contribute to multiple primary cancer. It required clinicians and researchers to explore a specific link between FPCs and SPCs. It would be of clinical importance to make a personalized screening plan for certain primary cancer patients (eg. oral pharyngeal cancer, ovary cancer and kidney cancer), especially for those young onsets. For example, patients with kidney cancer would have an increased risk of lung cancer and myeloma. Thus, a screening for lung cancer and myeloma should be recommended among those patients. Please refer to line 223-231.

Reviewer #1 (Recommendations for the authors):1. Four prominent groups of causal relationships mentioned in the research could be further explored. For instance, first primary ovary cancers can be classified into different subtypes to further analyze the relationship between subtypes and causal secondary primary cancers, so that secondary primary cancer prevention would be more specific and targeted.

Thank you for your comments. We performed subgroup analyses in SEER Research Plus Data 18 Reg, Nov 2020 Sub (2000-2018), with additional 2-year follow-up data. We stratified the total patients with first primary ovarian cancers to four groups based on the pathological subtypes: serous (~69.5%), endometrioid, mucinous, and clear cell carcinomas. The high-grade serous carcinoma (HGSC), as the most common type of ovarian cancer, was also evaluated. The SIR analyses were repeated in the above subgroups. The additional result was listed in supplementary file 1l. However, the number of patients with specific pathological type was too few to obtain a stable association result.

2. The limited SNPs included in some cancer types mentioned in the research might lead to missed opportunities to discover other significant associations. Enlargement of the data source could have benefits.

We totally agreed with you. Data sharing is critical to find significant associations. In particular, a complete genome-wide association study (GWAS) summary statistics that underpin the selection and weighting of genetic variants for a particular trait. Comprehensive databases of GWAS summary statistics, such as the pioneering NHGRI-EBI GWAS Catalog, are widely utilized by the community but still only a minority of published GWAS share their full summary statistics. The current study, however, provides the most up-to-date evidence in this topic. We also believe that it should be regularly updated with additional GWAS results in the future.

3. SEER database used in stage 1 and UK Biobank database used in stage 2 are based on completely different populations, and this may cause confounding factors due to genetic heterogeneity, which should be considered.

We mainly focus on the white people in SEER database (non-Hispanic whites) and UK Biobank (British, Irish and any other white background). As a matter of fact, since this study was mainly focus on the European Ancestry (aka, white or non-Hispanic or non-Nordic Caucasian), the genetic heterogeneity between these two populations was very low. [citation: https://handwiki.org/wiki/Social:Non-Hispanic_or_Latino_whites]

Reviewer #2 (Recommendations for the authors):The paper investigates the association and causal relationship of second primary cancers following a diagnosis of first primary cancer. The combination of a pan-cancer analysis and the use of Mendelian randomization adds novelty and value to the study. However, as the author state, the objective of the study is to investigate the causal relationship between FPCs and SPCs. This can be done based on the UKB data. it is not clear why use SEER data which only provides the association.

Thank you for being so concerned. The SEER database is the most significant incidence dataset in the United States based on population cancer registration. It is very critical to find significant associations between FPCs and SPCs based on such a great number of cancer patients. While the UKB data is composed of ~500,000 people aged 40-69 yrs, the incidence of FPCs and SPCs is limited, which might considerably impede the statistical power. On the other hand, in order to interpret the MR results under a more stringent standard, we would like to perform the analyses in independent study populations given the low genetic heterogeneity between these two populations. We think it is rigorous to combine these two stage results to draw a relatively solid conclusion.